# MicroRNA Expression Patterns Reveal a Role of the TGF-β Family Signaling in AML Chemo-Resistance

**DOI:** 10.3390/cancers15205086

**Published:** 2023-10-21

**Authors:** Paula Reichelt, Stephan Bernhart, Franziska Wilke, Sebastian Schwind, Michael Cross, Uwe Platzbecker, Gerhard Behre

**Affiliations:** 1Department of Hematology, Cell Therapy, Hemostaseology and Infectiology, University Hospital Leipzig, 04103 Leipzig, Germanymichael.cross@medizin.uni-leipzig.de (M.C.); uwe.platzbecker@medizin.uni-leipzig.de (U.P.); 2Interdisciplinary Center for Bioinformatics, Leipzig University, 04107 Leipzig, Germany; berni@bioinf.uni-leipzig.de; 3Dessau Medical Center, Clinic for Internal Medicine I—Gastroenterology, Hematology, Oncology, Hemostaseology, Palliative Medicine, Nephrology, Infectious Diseases, Pneumology, 06847 Dessau-Rosslau, Germany; gerhard.behre@klinikum-dessau.de

**Keywords:** acute myeloid leukemia, chemo-resistance, cytarabine, TGF-beta, activin signaling

## Abstract

**Simple Summary:**

In patients with acute myeloid leukemia (AML), cytarabine-based chemotherapy usually achieves remission, but this is commonly followed by relapse and chemo-resistance. In this study, we aim to establish next-generation sequencing (NGS)-based microRNA expression profiling and pathway analysis to identify pathways regulated differentially between chemo-sensitive and -resistant AML as potential therapeutic targets. MicroRNA expression profiles differ significantly between chemo-sensitive and chemo-resistant AML cells and reflect differences in the activity of intracellular signaling cascades. Alterations in signaling pathway activities contribute to treatment resistance and thus represent potential drug targets. Our microRNA-led approach indicates a role for activin receptor type 2A in ARA-C resistance of AML cells and suggests activin receptor signaling to be a candidate pathway for targeted therapy.

**Abstract:**

Resistance to chemotherapy is ultimately responsible for the majority of AML-related deaths, making the identification of resistance pathways a high priority. Transcriptomics approaches can be used to identify genes regulated at the level of transcription or mRNA stability but miss microRNA-mediated changes in translation, which are known to play a role in chemo-resistance. To address this, we compared miRNA profiles in paired chemo-sensitive and chemo-resistant subclones of HL60 cells and used a bioinformatics approach to predict affected pathways. From a total of 38 KEGG pathways implicated, TGF-β/activin family signaling was selected for further study. Chemo-resistant HL60 cells showed an increased TGF-β response but were not rendered chemo-sensitive by specific inhibitors. Differential pathway expression in primary AML samples was then investigated at the RNA level using publically available gene expression data in the TGCA database and by longitudinal analysis of pre- and post-resistance samples available from a limited number of patients. This confirmed differential expression and activity of the TGF-β family signaling pathway upon relapse and revealed that the expression of TGF-β and activin signaling genes at diagnosis was associated with overall survival. Our focus on a matched pair of cytarabine sensitive and resistant sublines to identify miRNAs that are associated specifically with resistance, coupled with the use of pathway analysis to rank predicted targets, has thus identified the activin/TGF-β signaling cascade as a potential target for overcoming resistance in AML.

## 1. Introduction

Standard chemotherapy regimens based on cytarabine (ARA-C) are effective at inducing remission in 70% of cases of acute myeloid leukemia (AML). However, the subsequent emergence of chemo-resistant clones often results in relapse, which is ultimately responsible for the majority of the 99,000 AML-related deaths occurring annually worldwide. The mechanisms that contribute to the evolution of chemo-resistance are highly variable between patients and remain poorly understood [1]. Their identification and characterization are a high priority since this knowledge should help to guide the use of existing therapies and identify potential targets for new ones.

MicroRNAs (miRNAs) affect gene expression post-transcriptionally by blocking translation and/or increasing turnover of their target mRNAs. A single miRNA typically targets several mRNAs, while each mRNA can be targeted by several miRNAs, creating a complex network of interactions that presumably helps coordinate metastable cell states. In this way, miRNAs can act as proto-oncogenes or tumor suppressors, depending on the cellular context [2,3,4]. Since the expression levels of certain miRNAs correlate with patient survival in a range of cancers, they are also potential prognostic markers [5,6,7,8].

We have previously identified multiple miRNAs involved in myelopoiesis and in the progression of AML [9,10,11,12,13,14] and have shown how a single miRNA can influence resistance to kinase inhibitors in FLT3-ITD-positive AML cells [15]. Indeed, there have been numerous reports of correlations between the expression level of specific miRNAs and drug resistance in a range of cancer types (reviewed in [16]), suggesting that miRNA expression patterns could be used to predict patient responses to specific therapy strategies. A study of eight independent AML cell lines with varying degrees of cytarabine sensitivity identified a number of miRNA/mRNA target pairs that are associated both with cytarabine resistance in vitro and with patient outcome [17]. Furthermore, it has been reported that the plasma levels of certain miRNAs change in AML patients undergoing chemotherapy [18]. As miRNAs can serve as mediators of crosstalk between signaling pathways, current studies focus both on individual miRNAs and on signatures coordinating pathway activity [19,20,21,22]. Indeed, miRNA signatures are likely to be more powerful than single miRNAs when it comes to predicting prognosis or identifying the pathways involved. Although these systems are complex, the growing potential of computational methods and the accumulation of relevant data in accessible databases are making their analysis increasingly feasible. In this study, we show how miRNA profiling can be used as a discovery tool to reveal dysregulated signaling pathways that contribute to drug resistance.

## 2. Materials and Methods

### 2.1. Cell Culture

Paired cell stocks from HL60 and ARA-C-resistant HL60R cells were kindly provided by Jindrich Cinatl (Institute of Medical Virology, Goethe-University, Frankfurt, Germany) [23]. Cells were cultured in parallel on RPMI containing 10% FCS and 1% penicillin/streptomycin/glutamine (PSG) and passaged every 48 to 72 h. To maintain resistance, the culture medium of HL60R cells was supplemented with cytarabine (ARA-C) to a final concentration of 8.25 µM.

### 2.2. miRNA Expression Profiling

In this study, 10 to 50 ng of total RNA was used in the small RNA protocol with the NEXTflex Small RNA-seq Kit v3 (Bioo Scientific, Austin, USA), according to the manufacturer’s instructions. A pool of 12 libraries was used for cluster generation at a concentration of 10 nM using an Illumina cBot. Sequencing of 50 bp was performed with an IlluminaHighScan-SQ sequencer at the sequencing core facility of IZKF Leipzig (Faculty of Medicine, University Leipzig) using version 3 chemistry and flow cell according to the manufacturer’s instructions.

### 2.3. Bioinformatics

Illumina small RNA adapters were clipped using cutadapt version 1.8.3, and sequencing quality was analyzed using fastqc (Andrews, S. (http://www.bioinformatics.babraham.ac.uk/projects/fastqc/, accessed on 1 July 2021)). Mapping of the clipped reads against the human hg38 assembly was performed using segemehl v 0.1.8. Reads mapping to miRNAs were counted using feature Counts with the -M-fraction options to count multiple mapping reads against the mature miRNAs of the human mirbase 21 annotation [24,25]. Differential expression was analyzed using edgeR, and all miRNAs with an adjusted *p* value < 0.05 and an absolute log2 fold change ≥ 1 were chosen for further analysis (Appendix A) [26,27]. We collected all known targets from mirTARbase (version July 2021) separately for miRNAs upregulated in parental and resistant HL60 cells [28]. Genes that were targets of miRNAs in both lists were removed, leaving 2775 genes in the HL60 and 1754 genes in the HL60R target set (Appendix A). KEGG pathways enriched for these target genes were then identified using DAVID [29]. The bioinformatic workflow is summarized in Appendix A.

### 2.4. Cell Proliferation Assays

Cells were seeded in 24-well plates at a density of 0.5 × 10^6^/mL and treated with concentrations of TGF-β between 0 and 10 ng/mL (Immunotools, Friesoythe, Germany). To check for synergism between TGF-β and ARA-C, HL60R cells were treated with 10 or 100 µM ARA-C in the presence or absence of 5 ng/mL TGF-β. Trypan-blue-stained cells were counted in a Neubauer chamber. Cell proliferation was also determined through the use of the MTS-assay (MTS Cell Proliferation Assay Kit/ab197010, abcam, Cambridge, UK) according to the manufacturer’s instructions.

### 2.5. Western Blotting

Cells were lysed in RIPA buffer containing 10 µM DTT (dithiothreitol, USBiological life science, Salem, MA, USA), 100 µM PMSF (phenylmethansulfonylfluorid, Sigma-Aldrich/Merck, Darmstadt, Germany) and 10 µL protease inhibitor cocktail (EDTA-Free, 100X in DMSO, Biotool, Kirchberg, Switzerland). Protein concentration was measured using a Bio-Rad protein assay kit with bovine serum albumin as standard.

Total cell extracts (20–50 μg) were resolved on a 10% sodium dodecyl sulfate–polyacrylamide gel and electroblotted to a PVDF membrane, which was then blocked with skimmed milk (5% in PBS-T). For protein detection, the following antibodies were used in a 1:1000 dilution: TGFBR1 polyclonal antibody (Elabscience, Houston, TX, USA), TGFBR2 antibody #79424 and Smad2/3 (D7G7) XP^®^ rabbit mAb #8685 (Cell Signaling, Danvers, MA, USA), ACVR2A E-AB-12278 (Elabscience, Houston, TX, USA) or 1:500 dilution: ACVR2B ab128544 (Abcam).

### 2.6. Flow Cytometry Assays

For intracellular staining of phosphorylated SMAD (pSMAD), cells were fixed in 1 mL of Lyse-Fix buffer (BD Bioscience, San Jose, CA, USA) for 10 min at 37 °C and then permeabilized on ice in permbuffer (BD Bioscience) for 30 min. Samples were washed with PBS containing 0.5% BSA and stained with 2 µL antibody per sample. For the assessment of inhibitor function, cells were seeded in 24-well plates at a density of 0.35 × 10^6^/mL and treated with 5 ng/mL TGF-β in the presence or absence of varying concentrations of inhibitors. Apoptosis was measured with a PE-AnnexinV/7AAD apoptosis detection kit (BD Bioscience) according to the manufacturer’s instructions after overnight treatment with TGF-β (5 ng/mL; Immunotools), galunisertib (3.5 µM) or SD208 (0.5 µM) (Selleckchem, Housten, TX, USA) in the presence or absence of 100 µM ARA-C. For cell cycle staining, 0.25 × 10^6^ pelleted cells were resuspended in 250 µL PBS/EDTA (1 mM). Samples were fixed by the addition of 750 µL absolute ethanol and overnight incubation at 4 °C. Prior to staining, cells were washed in PBS and treated with 100 µg/mL RNAse A in a total volume of 475 µL PBS/EDTA for 20 min at 37 °C. Propidium iodide (Sigma) was added to a final concentration of 50 µg/mL to distinguish between cell cycle phases in terms of cellular DNA content.

### 2.7. Patient Samples and Quantitative Real-Time PCR

Bone marrow samples from four AML patients were obtained from the University of Leipzig Medical Center after informed consent as approved by the local ethics committee. The patient characteristics are presented in Appendix A. Frozen bone marrow samples (lysed in GC-Buffer) were thawed and subjected to RNA isolation using a NucleoSpin miRNA kit (MACHEREY-NAGEL, Düren, Germany) according to the manufacturer’s instructions. Primer design for all mRNA targets was performed using NCBI primer blast, Primer3plus and UCSC tools. A list of primer sequences is provided in Appendix A.

### 2.8. Statistical Analyses

Experimental data are expressed as mean ± SD. Statistical significance between control and treated groups was determined using Student’s *t*-test when data were normally distributed. *p* values < 0.05 and <0.01 were considered significant and highly significant, respectively. For TCGA database analysis, patients were subdivided by risk groups and mRNA expression data were tested for normalcy and equal variance using Shapiro–Wilk and Levene tests. Finally, significant differences within the groups were analyzed using Kruskal–Wallis and Dunn test. For overall survival plots, patients were subdivided into high and low expression by their median. Kaplan–Meier plots and statistics were generated using the survminer package in R studio software (R version 3.6.1 x86_64-w64-mingw32/x64 (64-bit)) [30].

## 3. Results

### 3.1. MiRNA Profiling Suggests the Involvement of TGF-β Family Signaling in Chemo-Resistance

MiRNA expression profiling identified 24 miRNAs to be expressed differentially (*p* < 0.05) between ARA-C sensitive and ARA-C resistant HL60 cells (Figure 1). A bioinformatics analysis that first identified predicted target mRNAs and then assessed their representation in KEGG (Kyoto Encyclopedia of Genes and Genomes) [31] implicated 38 pathways to be potentially involved in chemo-resistance (false discovery rate, FDR < 0.05). The pathways predicted to be the most affected are shown ranked by FDR in Table 1. Four of the ten highest ranked pathways are receptor-mediated signaling pathways (erbB, neurotrophin, TGF-β, and T cell receptor) and of these, it was decided to focus on the TGF-β family pathway firstly because of its known relevance to hematopoiesis, where it tends to inhibit proliferation while stimulating differentiation and apoptosis [32], and secondly because the pharmacologic neutralization of TGF-β1 has been previously observed to enhance ARA-C-induced apoptosis in AML cells [33]. The relevant miRNAs that are either increased or decreased in chemo-resistance are shown together with their predicted target mRNAs in Table 2.

Among the predicted targets, the TGF-β family receptors show a consistent picture, with TGFBR1 and ACVR2A being predicted targets for two and three separate miRNAs, respectively, which are all decreased in the resistant cell line, while ACVR2B is predicted to be targeted by three separate miRNAs increased in the resistant cell line. Should the expressed miRNAs indeed have a decisive influence on protein expression, this pattern predicts higher levels of TGFBR1 and ACVR2A in the chemo-resistant HL60R cells and higher levels of ACVR2B in the chemo-sensitive HL60 cells.

### 3.2. Chemo-Resistant Cells Have Altered Levels of TGF-β Signaling Proteins and Their Targets

Protein analysis through the use of Western blotting confirmed that both TGFBR1 and TGFBR2 were present at significantly higher levels in the resistant cells than in the chemo-sensitive cells from which they derive (Figure 2A,B). The pattern of ACVR2 protein expression was also consistent with the miRNA pattern, with the resistant line showing a significant increase in ACVR2A and an accompanying decrease in ACVR2B (Figure 2C,D). A higher overall level of TGF-β-family signaling in the chemo-resistant cell line is further supported by the increased expression of the signal-transducing protein SMAD2 (Figure 2F)— increased phosphorylation of SMAD2/3 (Figure 2G) and the decreased level of cMYC (Figure 2E)—the expression of which is known to be negatively regulated by TGF-β.

Taken together, these data confirm that the acquisition of cytarabine resistance by HL60 cells is accompanied by significant changes in the expression of TGF-β family pathway proteins, as predicted through the use of miRNA profiling.

### 3.3. TGF-β Receptor-Mediated Signaling Affects the Proliferation of Chemo-Resistant Cells, but Is Not a Major Determinant of Chemo-Resistance

The upregulation of the TGF-β family receptor and signaling proteins in chemo-resistant HL60 cells suggests that a response to cognate ligands present in the serum-containing medium may contribute to chemo-resistance in this model. As ARA-C interferes with DNA replication during the synthesis phase, a TGF-β signaling-mediated decrease in cycling rate might support resistant cells by allowing more time for DNA damage response. To test for a link between TGF-β signaling and chemo-resistance, we first tested the effects of recombinant human TGF-β on cell proliferation, as assessed through the use of cell counting and MTS assay. Three days of TGF-β treatment resulted in dose-dependent growth inhibition of the chemo-resistant HL60R cells without affecting the chemo-sensitive HL60 cells (Figure 3A,B and Appendix A), which is consistent with increased receptor signaling capacity in the resistant cells.

However, the reduction in proliferation rate in 5 ng/mL TGF-β was not accompanied by increased resistance to ARA-C (Figure 3C). To examine this in more detail, cultures were treated with combinations of TGF-β and ARA-C and subject to periodic analysis of both cell cycle and apoptosis (Figure 4 and Appendix A). This showed that TGF-β treatment caused only a modest accumulation of cells in the G0/G1 phase of the cycle, with around 40% of the cells still in S or G2. The TGF-β dependent reduction in cell yield, therefore, appears to be indicative not of a static culture but of a dynamic state in which cycling is balanced by apoptosis. While the inclusion of ARA-C to 10 µM had little or no effect on cell cycle distribution, 100µM ARA-C led to an accumulation of cells in the S and G2/M phases, probably as a consequence of extensive DNA damage and prolongation of these phases due to increased repair processes. Importantly, the inclusion of TGF-β during ARA-C treatment tended to increase rather than decrease the proportion of cells undergoing apoptosis (Figure 4A,B).

The failure of additional TGF-β to increase chemo-resistance does not necessarily rule out a chemo-protective effect of TGFBR1 ligands that may be present in the serum-containing growth medium. To test the relevance of this background TGFBR1 signaling, we used the TGFBR1 receptor inhibitors SD208 and galunisertib. After confirmation of the concentration-dependent inhibition of SMAD2/3 phosphorylation using both galusinertib and SD208 (Appendix A), the cells were subjected to treatment with each inhibitor alone and in combination with 100 µM ARA-C. Neither of the TGFBR1 inhibitors increased the overall proliferation rate of the HL60R cells (Figure 4C), and while each inhibitor did tend to increase ARA-C-dependent apoptosis measured after 24 h, this effect was relatively small and did not reach significance (Figure 4D).

The effects of specifically blocking TGFBR1 were, therefore, similar to those of TGF-β supplementation. This means that any role for TGF-β family signaling in mediating chemo-resistance in the cell model is highly likely to involve ligand–receptor combinations other than TGF-β/TGFBR1.

### 3.4. Changes in TGF-β Pathway Expression Associated with the Development of Resistance in Primary AML

The experiments reported above imply a role for TGF-β family signaling, but not necessarily for TGF-β itself, in ARA-C resistance emerging under selection in a leukemic cell line. Although these changes were originally identified via extrapolation from differential miRNA expression patterns, changes in pathway components were also noted at the mRNA level, most clearly for the activin receptors ACVR2A and ACVR2B and the target of TGF-β family signaling CDKN2B (Figure 5A). In order to extend these studies to a more clinically relevant system, we, therefore, isolated RNA from paired bone marrow samples taken from four individual AML patients at diagnosis and at relapse and analyzed the mRNA expression levels of a selection of signaling and target genes of the TGF-β/activin pathway (Figure 5B). All patients had received standard chemotherapy without experimental treatments or allogenic stem cell transplantation between the tested time points of diagnosis and relapse. All four patients had relapsed after comparatively short time periods of 113 to 266 days despite an initially favorable or intermediate prognosis. Patient characteristics are summarized in Appendix A.

Consistent with the HL60/HL60R cell line model, the development of chemo-resistance in vivo was accompanied by a clear and consistent increase in mRNA levels of both the activin receptor *ACVR2A* and the TGF-β family target *CDKN2B,* the expression of which is activated by TGF-β family signaling and which was included here as a reporter of pathway activity [34] (Figure 5B). Similarly, the reduction in c*MYC* mRNA, while not reaching significance, is consistent with the reported negative regulation of *cMYC* by TGF-β-related signaling [35]. The primary AML samples taken after the relapse also had a small but significant increase in TGFBR1 mRNA and a trend toward higher TGFBR2 mRNA. This represents a difference to the cell line model, in which the upregulation of these gene products was apparent at the protein but not at the RNA level (Figure 2 and Figure 5A).

Although this longitudinal analysis of AML patients at diagnosis and relapse is limited to four patients, it confirms that the acquisition of chemo-resistance in vivo is accompanied by increases in TGF-β pathway gene expression.

### 3.5. TGF-β Family Pathways Are Implicated in the Prognosis and Chemo-Resistance of Primary AML

Based on the differential expression of the TGF-β family signaling components in the relapsed AML samples, we went on to compare the mRNA expression levels of TGF-β signaling components between the AML risk groups using the comprehensive data available in the cancer genome atlas (TGCA) database [36].

This revealed the signal transducers SMAD2 and SMAD3 to be more highly expressed in the poor risk than in the favorable risk group (Figure 6A,B). High expression of SMAD3 and TGFB1 also correlated with decreased overall survival (Figure 6G,I). Furthermore, higher expression of SMAD6 and SMURF1, which are negative regulators, correlated with a more favorable prognosis and increased overall survival, respectively (Figure 6C,H). 

The expression of the TGF-β receptors TGFBR1 and 2 did not differ significantly between the AML risk groups. However, the expression of TGFB1 itself was increased in both the intermediate and high risk compared to the favorable risk group (Figure 6E). Most notably, the high-risk group also had significantly increased expression of both activin A (INHBA, Figure 6F) and the receptor ACVR2A (Figure 6D).

Given our observation of a reciprocal pattern of ACVR2A and ACVR2B protein expression between chemo-sensitive and chemo-resistant HL60 cells (Figure 2) and the increase in ACVR2A mRNA seen upon relapse of primary AML (Figure 5), it was of particular interest to test for correlation between mRNA expression levels and overall survival in the TGCA dataset. Interestingly, while neither ACVR2A nor ACVR2B expression alone correlated with overall survival, the ratio of ACVR2A to ACVR2B expression clearly did, with a high ratio (high expression of ACVR2A) being associated with significantly shorter overall survival (Figure 7).

## 4. Discussion

We show here that miRNA profiling in an AML cell line model of induced chemo-resistance can be used to identify pathways that are relevant to resistance and prognosis in vivo. The rationale behind using miRNAs to predict differentially regulated pathways is their role in post-transcriptional regulation. Indeed, the majority of the predicted targets identified in this way and then tested at both mRNA and protein levels in the HL60 cell lines showed significant changes only at the protein level. On this basis, these would not have been picked up during RNA sequencing.

Candidate pathways of potential relevance to chemo-resistance were identified in silico based purely on the representation of predicted miRNA targets. For the purpose of this study, the TGF-β family pathway was chosen for closer analysis firstly because it is of known relevance to hematopoiesis, secondly, because it is a receptor-mediated signaling pathway open to selective inhibition and thirdly because the miRNA expression pattern suggested that at least some targets were induced, rather than suppressed in the resistant state. The other candidate pathways identified are clearly also of potential relevance, and it will be interesting to study these in more detail in the future.

A targeted analysis of gene expression in primary AMLs pre- and post-resistance validated the TGF-β family targets identified in the cell line model. In some cases, there were changes in mRNA levels in the primary samples but not in the cell line model, where regulation was purely post-transcriptional. Although mRNA levels in patient samples were determined across the whole populations of bone marrow mononuclear cells and do not necessarily reflect changes within single cell types, this does demonstrate that resistance genes may be activated both transcriptionally and post-transcriptionally, depending on the circumstances.

The fact that the HL60R but not the HL60 cells reduced proliferation in response to TGF-β confirms that resistance is indeed accompanied by functional changes in the TGF-β signaling pathway. However, TGF-β supplementation provided no extra protection against ARA-C and actually appeared to increase the rate of apoptosis. This suggests either that the increased signaling capacity is not relevant for chemo-resistance or that ligands already present in the medium offer maximum protection and that supplemented TGF-β increases other aspects of the response. Indeed, the TGF-β response is known to be multifaceted and highly context-dependent and to include an increase in apoptosis [37]. In this case, the ligand responsible for resistance is unlikely to be TGF-β itself since specific inhibition of TGF-β signaling with the TGFBR1 kinase inhibitors galunisertib or SD208 reduced SMAD phosphorylation but had little or no effect on either proliferation rate or chemo-sensitivity.

A number of observations suggest a possible role for activins in this respect. Firstly, the activin receptor 2 isotypes ACVR2A and ACVR2B were both identified as targets of differentially expressed miRNAs; secondly, we found these to be expressed reciprocally at both the protein and mRNA level in the chemo-sensitive HL60 and chemo-resistant HL60R cells, with high ACVR2A in the latter; thirdly, our targeted analysis of gene expression in the publically available TGCA database showed that poor-risk patients expressed higher levels of both ACVR2A and the ligand activin A (INHBA); and finally, we found that the ratio of ACVR2A to ACVR2B expression is associated with overall survival across all AML. This confirms and extends the results of a previous study that identified the expression of the ligand activin A to be similarly associated with poor response to therapy and poor survival [38]. The fact that the ratio of ACVR2A to ACVR2B mRNA is associated with overall survival, but not the expression of either alone, is suggestive of interference between receptor complexes, either in terms of competition for binding to the ligand or the common ACVR1 receptor subunit, or at the level of downstream signaling. It will be interesting to characterize these interactions in more detail as potential targets for therapy.

Detailed analysis of gene expression changes accompanying the transition from chemo-sensitive to chemo-resistant AML is limited by the low number of appropriate longitudinal samples since patients who are treated exclusively with standard of care tend to undergo stem cell transplantation during remission. However, despite the small sample size, our targeted analysis of gene expression in four individual patients progressing from an initial chemo-sensitive to a chemo-resistant state confirmed significant overexpression of both TGFBR1 and ACVR2A as well as the TGF-β target gene CDK2NB in the resistant state. As noted above, TGF-β family signaling networks are highly complex, interactive and context-dependent, making it difficult to predict the effects of any one change. However, activin signaling has been found to induce G1 arrest in a variety of situations and, therefore, has the potential to contribute to chemo-resistance, as described above [39]. While no selective activin receptor inhibitors are currently available, ligand trap approaches are already in clinical use and may, in the future, provide a means of selective targeting ACVR2A or B receptor signaling [40,41,42].

## 5. Conclusions

In summary, we show here how the use of computational tools can imply changes in pathway activities from miRNA expression profiles, leading in this case to the identification of activin/TGF-β family signaling as a potential mediator of chemo-resistance both in a cell line model and in primary AML. It is important to note that this provides proof of principle for the chosen approach rather than evidence of a role for miRNAs in chemo-resistance in vivo. The clarification of chemo-resistance mechanisms based on this approach promises to provide valuable information both on potential targets for therapy and on patient-specific characteristics of disease that are expected to become increasingly relevant to personalized therapy.

## Figures and Tables

**Figure 1 cancers-15-05086-f001:**
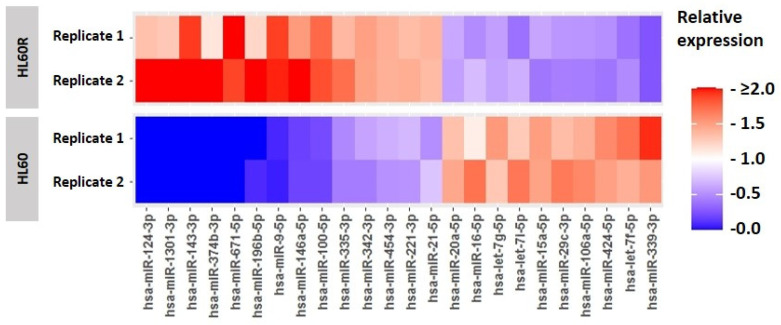
MicroRNA profiling. MicroRNA expression in cells resistant (HL60R) and sensitive (HL60) to ARA-C were determined via next-generation sequencing (NGS) of small RNAs. Data from 2 technical replicates for each subline are presented as expression level relative to the mean of all four levels measured for each microRNA.

**Figure 2 cancers-15-05086-f002:**
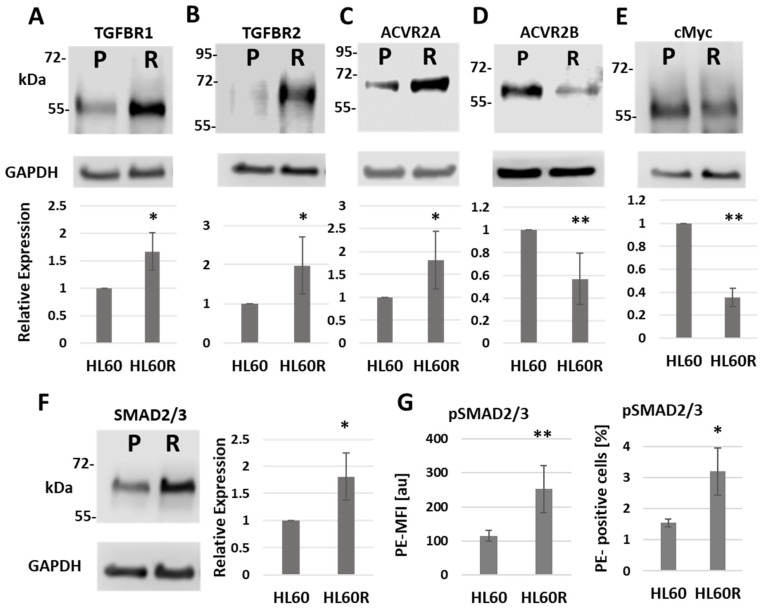
ARA-C-resistant cells show higher expression levels of TGF-β family signaling proteins and increased basal SMAD phosphorylation. Representative Western blots are shown together with the corresponding relative expression levels of (**A**) TGF-β-Receptor (TGFBR) I (*n* = 4), (**B**) TGFBRII (*n* = 4), (**C**) Activin-Receptor (ACVR) IIA (*n* = 6), (**D**) ACVRIIB (*n* = 6), (**E**) c-Myc (*n* = 3) and (**F**) SMAD2/3 (*n* = 3) in parental (P/HL60) and resistant (R/HL60R) cells. (**G**) Basal phosphorylation level of SMAD2/3 in parental and chemo-resistant cells assessed via phosflow cytometry: left, mean florescence intensity (MFI) of PE in arbitrary units (au); right, percentage of PE (pSMAD)-positive cells. *p*-values were calculated using paired Student’s *t*-test. Statistical significance is indicated as * *p* ≤ 0.05 and ** *p* ≤ 0.01. Original western blots are presented in Appendix A.

**Figure 3 cancers-15-05086-f003:**
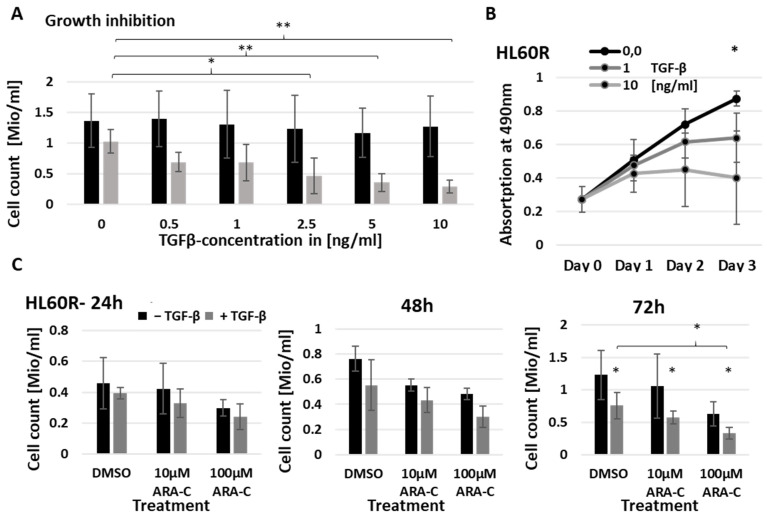
Chemo-resistant cells show an increased response to TGF-β treatment. (**A**) cell counts of chemo-sensitive HL60 and chemo-resistant HL60R cells after 3 days of treatment with increasing concentrations of TGF-β (*n* = 3); (**B**) growth curve of HL60R cells treated with 0; 1 and 10 ng TGF-β, assessed by MTS assay (*n* = 3); (**C**) HL60R cell counts after 24, 48 and 72 h of treatment with combinations of ARA-C and 5 ng/mL TGF-β (*n* = 3). *p*-values were calculated using paired Student’s *t*-test. Statistical significance is indicated as * *p* ≤ 0.05 and ** *p* ≤ 0.01.

**Figure 4 cancers-15-05086-f004:**
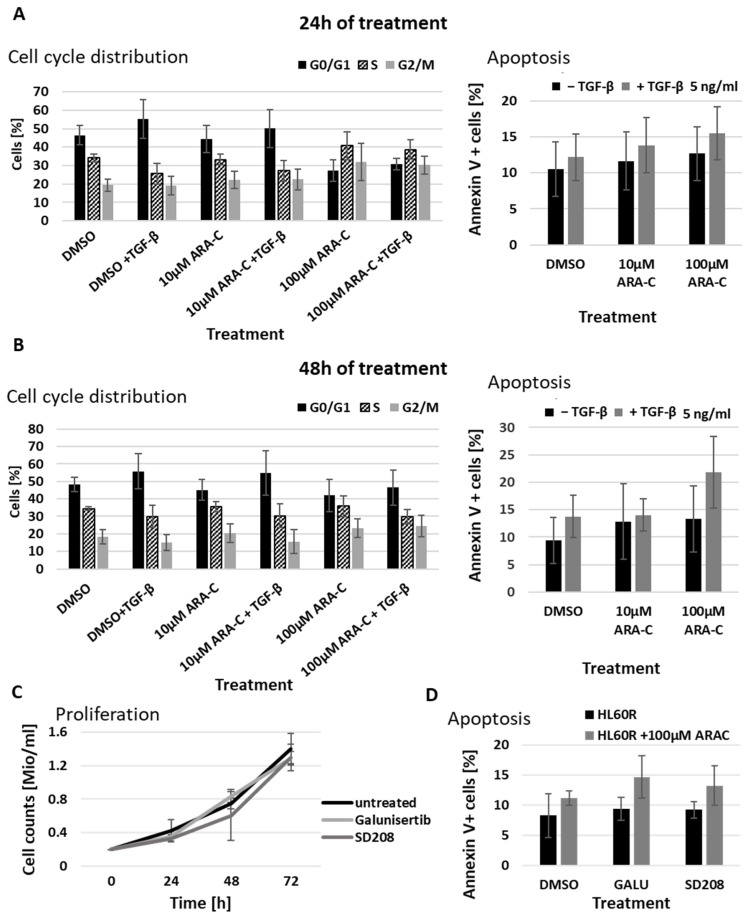
Increased TGF-β signaling does not mediate chemo-resistance. (**A**,**B**) Distribution of HL60R cells over cell cycle phases (**left**) and apoptosis rates (**right**) after combined or single treatment with TGF-β and increasing concentrations of ARA-C for (**A**) 24 h and (**B**) 48 h; (*n* = 4) (**C**) Analysis of proliferation rate in ARA-C resistant HL60R cells by treatment with TGF-β signaling inhibitors SD208 and Galunisertib (GALU). (**D**) Analysis of apoptosis induction in HL60R cells by combination of ARA-C treatment with TGFβ signaling inhibitors SD208 and Galunisertib (GALU).

**Figure 5 cancers-15-05086-f005:**
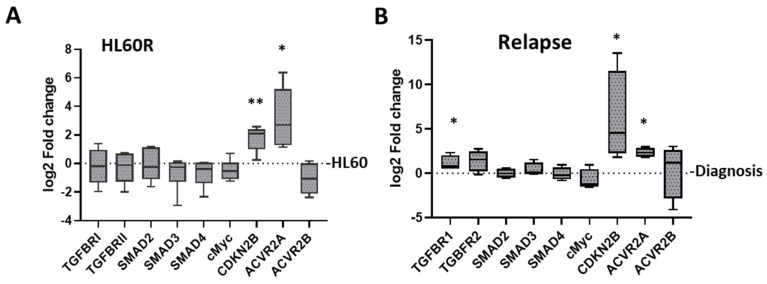
Chemo-resistance and relapse are associated with upregulation of TGF-β/Activin signaling genes. (**A**) Gene expression in ARA-C resistant HL60R cells as log fold change compared to expression in the chemo-sensitive HL60 cells. Data are shown from six independent replicates. (**B**) Expression of selected TGF-β family pathway genes in relapsed AML patients as log fold change compared to the expression levels at diagnosis. Center lines show the medians; box limits indicate interquartile range and whiskers extend to minimum and maximum values. *p*-values were calculated using paired Student’s *t*-test for normally distributed and Dunn test for abnormally distributed data (CDKN2B and TGFBR1). Statistical significance is indicated as * *p* ≤ 0.05 and ** *p* ≤ 0.01.

**Figure 6 cancers-15-05086-f006:**
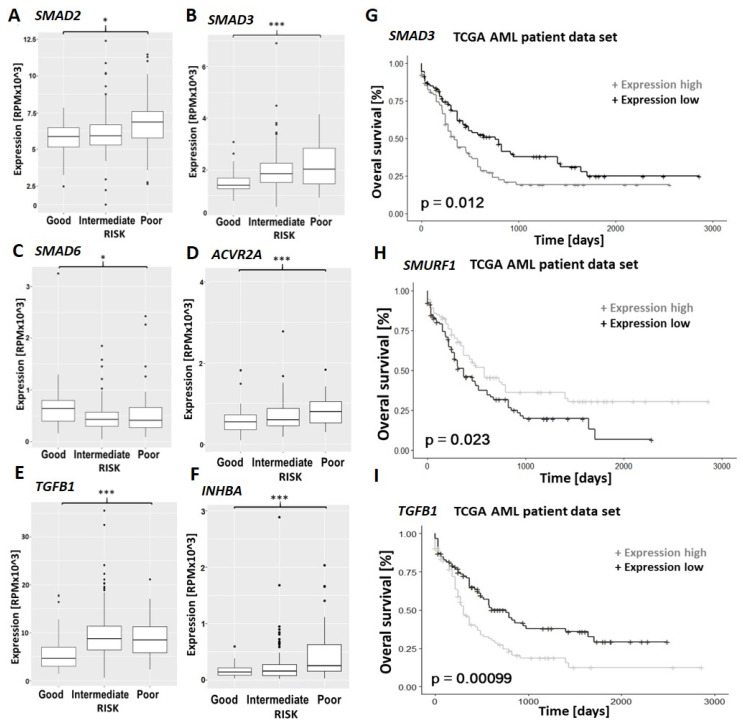
TGF-β signaling proteins are prognostic factors in AML. Expression of (**A**) SMAD2, (**B**) SMAD3, (**C**) SMAD6, (**D**) Activin Receptor IIA, (**E**) TGF-β1 and (**F**) Activin A (reads per million) in AML risk groups; (**G**–**I**) Overall survival of AML patients according to SMAD3, SMURF1and TGF-β expression above or below the median. *p* values were calculated using a Log-rank test (*n* = 198). Statistical significance is indicated as * *p* ≤ 0.05 and *** *p* ≤ 0.001.

**Figure 7 cancers-15-05086-f007:**
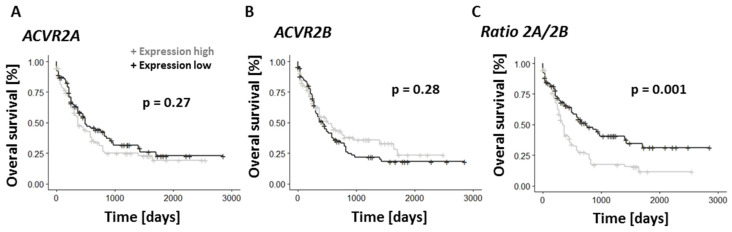
Ratio of activin receptor 2A/2B expression has prognostic significance in AML. Overall survival rates of AML patients according to (**A**) ACVR2A expression (**B**) ACVR2B expression, and (**C**) the ratio of expression ACVR2A/2B above or below the median, using data from the TCGA AML patient dataset (*n* = 198). *p*-values were calculated using Log-rank test.

**Table 1 cancers-15-05086-t001:** Signaling pathways predicted via bioinfomatic evaluation of microRNA profiles ranked by FDR.

Term	Count	% Coverage	FDR
Pathways in cancer	186	4.1	5.89 × 10^−5^
Proteoglycans in cancer	81	1.78	0.00104
Cell cycle	54	1.19	0.0016
ErbB signaling pathway	40	0.88	0.0016
Regulation of actin cytoskeleton	83	1.83	0.0016
Neurotrophin signaling pathway	51	1.12	0.0019
TGF-beta signaling pathway	42	0.92	0.0029
T cell receptor signaling pathway	45	0.99	0.0034
Salmonella infection	89	1.96	0.0063
MAPK signaling pathway	102	2.25	0.0067
Hippo signaling pathway	60	1.32	0.0096
Protein processing in endoplasmic reticulum	64	1.41	0.0106
Renal cell carcinoma	31	0.68	0.0152
Lipid and atherosclerosis	76	1.67	0.0162
EGFR tyrosine kinase inhibitor resistance	34	0.75	0.0162

**Table 2 cancers-15-05086-t002:** MicroRNAs and predicted targets within the KEGG pathway TGF-β signaling.

miRs Upregulated in HL60R	Targets	miRs Downregulated in HL60R	Targets
hsa-miR-100-5p	CREBBP	hsa-let-7f-5p	TGFBR1
	CUL1		THBS1
	ID1	hsa-let-7g-5p	TGFBR1
	BMP6		THBS1
	BMPR1A	hsa-let-7i-5p	BMP4
	E2F4		THBS1
	FMOD		BMP2
hsa-miR-124-3p	GREM1	hsa-miR-106a-5p	BMP8B
	ID1		RGMB
	ID2		ACVR2A
	ID4		CDKN2B
	RHOA	hsa-miR-15a-5p	IFNG
	ROCK1		RPS6KB1
	ACVR2B		SMURF1
hsa-miR-1301-3p	CREBBP		ACVR2A
	SKP1		BAMBI
hsa-miR-143-3p	TNF		IFNG
hsa-miR-146a-5p	RHOA	hsa-miR-16-5p	RPS6KB1
	ROCK1		SMAD1
hsa-miR-196b-5p	ACVR2B		SMURF1
	FMOD		SMURF2
	GDF5	hsa-miR-20a-5p	BAMBI
hsa-miR-21-5p	TGFB2		BMP2
	TGIF1		BMP8B
	ZFYVE16		RBL1
hsa-miR-221-3p	ACVR2B		RGMB
	RHOA		SMAD6
hsa-miR-335-3p	ID2		TGFBR1
hsa-miR-342-3p	BMP7		THBS1
	ID4	hsa-miR-29c-3p	FBN1
hsa-miR-374b-3p	TGIF1		TGIF2
hsa-miR-454-3p	ACVR1	hsa-miR-424-5p	ACVR2A
hsa-miR-671-5p	BMP8A		RPS6KB1
	PPP2CA		SMURF1

## Data Availability

The materials described in the manuscript, including all relevant raw data, will be freely available to any researcher wishing to use them for non-commercial purposes, without breaching participant confidentiality. Datasets generated during and/or analyzed during the current study are available from the corresponding author upon reasonable request.

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
