# Peer review of "MicroRNA Expression Patterns Reveal a Role of the TGF-β Family Signaling in AML Chemo-Resistance"

_cancers, 2023, doi:10.3390/cancers15205086_

Round 1

Reviewer 1 Report

The authors described the involvement of TGF-beta pathway and TGF-beta-associated miRNAs in resistance to ara-C treatment of AML cell lines. The manuscript is well written, however, there are some issues to be addressed to prior to manuscript acceptance.

-          All the statistical results throughout the manuscript should be recalculated and presented as mean +/- SD, not SEM. Standard error of mean is not a measure of data dispersion and variability and as such it should not be used as a part of descriptive statistics

-          The authors should list all differentially expressed miRNAs after RNA-seq analysis, together with fold change and FDR values. Also, all miRNA targeted genes in both cell lines (that is 895 genes in HL60 and 558 genes in HL60R) should be listed. Both lists can be added as a supplementary table or supplementary Excel file.

-          Where are the miRNAs from Table 2 from? It seems that they are not a part of the 24 differentially expressed miRNAs discovered after RNA-seq.

-          Figure 2 legend: there is some disorder in the explanation of MFI and PE

Reviewer 2 Report

The submitted manuscript is well-written and well-presented, covering an interesting point on profiling miRNA signatures in cytarabine-sensitive and -resistant AML and suggesting the dysregulated signaling pathways contributing to cytarabine resistance.

However, there are some comments that I would like to share with the authors; 

- As there are several studies investigating the involvement of miRNAs in the acquired resistance of leukemia cells to cytarabine chemotherapy, I would recommend adding a significant/novelty statement following the abstract to clarify the added value of this work. 

- Several previous studies that contribute significantly to the same point are not covered or discussed in the present study. For example, the study by Konopleva et al. 2013, is one of the earliest studies to discover that blocking TGFβ could enhance cytarabine-induced apoptosis in AML cells.  Also, the study by Jatinder K Lamba in 2015, who performed genome-wide miRNA profiling of AML cell lines to identify miRNAs associated with cytarabine chemo-sensitivity, is another study that shares the same interest but not cited or discussed in the present study.

- The authors stated that “ miRNA expression pattern suggested that there are qualitative changes in signaling between sensitive and resistant states.” It is not clear what is meant by “qualitative changes”?

- In the Flow cytometry experiment for cell cycle analysis and apoptosis analysis, I have some comments;

   1- In the flow chart, the G0/G1 phase was not represented, and the G2/M phase was not mentioned in the analysis of the different cycles.

  2- The term “G0/G1” is usually used, not the opposite, as stated in line 256.

  3 - This part was not covered well or thoroughly discussed. No clear explanation was offered to explain why treating cells with 10 ul ARA-C (with or without TGFβ) induced cell arrest at G1, but for 100 ul ARA-C, the cell arrest was detected mainly at G2?

  4 - The title of Supplementary Figure 3 ”Induction of G1 cell cycle arrest by TGFβ in combination with different concentrations of ARA-C” is not accurate. As it can be noticed that the cell cycle arrest was induced at the G2 when TGFβ is combined to 100 ul ARA-C 

  5- For apoptosis analysis, the chart of early and late apoptosis with the four quadrants should be submitted and added to the manuscript.

- For gene analysis in patient samples, the sample size is not indicative and is insufficient for reliable results.

- Although SMURF1 is a predicted target for three separate miRNAs, it was not selected for further analysis.

 In conclusion, this is an exciting paper suitable for publication in the journal but after a good revision.
